# Novel Input for Designing Patient-Tailored Pulmonary Rehabilitation: Telemonitoring Physical Activity as a Vital Sign—SMARTREAB Study

**DOI:** 10.3390/jcm9082450

**Published:** 2020-07-31

**Authors:** Catarina Duarte Santos, Rui César das Neves, Ruy M. Ribeiro, Cátia Caneiras, Fátima Rodrigues, Martijn A. Spruit, Cristina Bárbara

**Affiliations:** 1Instituto de Saúde Ambiental (ISAMB), Faculdade de Medicina, Universidade de Lisboa, 1649 028 Lisbon, Portugal; fatima.pneumo@gmail.com (F.R.); cristina.barbara@chln.min-saude.pt (C.B.); 2Unidade de Reabilitação Respiratória, Hospital Pulido Valente, Centro Hospitalar Universitário Lisboa Norte, 1769 001 Lisbon, Portugal; 3CAST—Consultoria e Aplicações em Sistemas e Tecnologia, Lda., 1800 075 Lisbon, Portugal; rcneves@cast.pt; 4Laboratório de Biomatemática, Instituto de Saúde Ambiental (ISAMB), Faculdade de Medicina, Universidade de Lisboa, 1649 028 Lisbon, Portugal; ruyribeiro@medicina.ulisboa.pt; 5Laboratório de Microbiologia na Saúde Ambiental (EnviHealthMicroLab), Instituto de Saúde Ambiental (ISAMB), Faculdade de Medicina, Universidade de Lisboa, 1649 028 Lisbon, Portugal; ccaneiras@gmail.com; 6Healthcare Department, Nippon Gases Portugal, 4470 177 Maia, Portugal; 7Department of Research & Development, Centre of Expertise for Chronic Organ Failure (CIRO), 6085 NM Horn, The Netherlands; martijnspruit@ciro-horn.nl; 8Department of Respiratory Medicine, Maastricht University Medical Centre, NUTRIM School of Nutrition and Translational Research in Metabolism, Faculty of Health, Medicine and Life Sciences, 6229 HX Maastricht, The Netherlands; 9REVAL-Rehabilitation Research Center, BIOMED-Biomedical Research Institute, Faculty of Rehabilitation Sciences, Hasselt University, 3590 BE Diepenbeek, Belgium; 10Serviço de Pneumologia, Centro Hospitalar Universitário Lisboa Norte, 1649 028 Lisbon, Portugal

**Keywords:** physical activity, telemonitoring, vital sign, respiratory diseases, IPAQ

## Abstract

Physical inactivity may be a consequence of chronic diseases but also a potential modifiable risk factor. Therefore, it should be clinically assessed as a vital sign of patients’ general physical condition prior to any exercise-based intervention. This cross-sectional study describes physical activity in the daily life of 100 chronic respiratory patients before pulmonary rehabilitation, comparing subjective and objective measures. The assessment combined the International Physical Activity Questionnaire (IPAQ) and 4-day accelerometer and oximeter telemonitoring with SMARTREAB technology, assessing heart rate, transcutaneous oxygen saturation and activity-related energy expenditure by metabolic equivalent of task (MET). According to IPAQ, 49% of patients had a moderate level of physical activity in daily life (PADL), a weekly mean level of 2844 ± 2925 MET.min/week, and a mean sedentary time of 5.8 ± 2.7 h/day. Alongside this, SMARTREAB telemonitoring assessed maximum activity ranging from 1.51 to 4.64 METs, with 99.6% daytime spent on PADL below 3 METs and 93% of patients with daily desaturation episodes. Regardless of the self-reported IPAQ, patients spend at least 70% of daytime on PADL below 2 METs. SMARTREAB was demonstrated to be an innovative methodology to measure PADL as a vital sign, combining oximetry with accelerometry, crossmatched with qualitative patient data, providing important input for designing patient-tailored pulmonary rehabilitation.

## 1. Introduction

Physical inactivity is the 4th worldwide risk factor for ill health with 1 million deaths and 8.3 million disability-adjusted life years lost per year [1]. Acknowledging this, the initiative Exercise is Medicine^®^ encourages all healthcare professionals to promptly assess physical activity habits as a vital sign of general physical condition, optimizing exercise counselling or referral and enhancing chronic disease management [2].

Although 10,000 steps/day is a reasonable target for healthy adults who can take between 4000 and 18,000 steps/day [3], such values do not apply to patients with chronic respiratory disease. Chronic obstructive pulmonary disease (COPD) patients have less walking time at a slower walk pace [4,5], with a mean of almost 5000 steps/day [6,7] and associated increased risks for exacerbation-related hospitalization and premature mortality [8,9]. On average, all major subtypes of interstitial lung disease (ILD) patients have reduced physical activity with persistent pain and psychological deficits [10], and an annual decline of nearly 1000 steps per day, which is twice as high as compared to COPD patients [11]. About 90% of bronchiectasis patients do not meet the recommended physical activity guidelines and have more than 10 h of daily sedentary behavior, walking an average of 6000 steps/day [12]. Daily physical activity is also decreased on asthma patients with reported 8546 steps/day for mild/moderate asthma and 6560 steps/day for severe asthma [13]. 

The average physical activity level measured by the number of daily steps may not be the most appropriate measure to apply as a vital sign of general physical condition in patients with a chronic respiratory disease, given that there is a great variability in patients’ physical activity patterns [5,14]. Because of this, research on pulmonary rehabilitation targets development of methodology for physical activity assessment, combining subjective patient reported experience and accurate objective measurement on patients’ daily life [4,15]. 

In 2009 our pulmonary rehabilitation Unit started TELEMOLD, a project which developed an Android smartphone telemonitoring system with a mobile software application connected to oximeter and accelerometer sensors. This innovative telemedicine solution was validated on healthy subjects and later applied to COPD patients for optimization of long-term oxygen therapy prescription [16]. Currently the project has been upgraded to SMARTREAB with physical activity in daily life (PADL) being measured as a vital sign of patients’ general physical condition. Used in clinical routine with chronic respiratory patients, the crossmatching of a combined oximetry and accelerometry with patient qualitative input has been important on designing an individually-tailored pulmonary rehabilitation.

The aims of this study were: (1) to compare PADL subjective and objective measures in patients with chronic respiratory diseases; and (2) to demonstrate how SMARTREAB provides important input for designing patient-tailored pulmonary rehabilitation.

## 2. Experimental Section

Patients with various chronic respiratory diseases referred for pulmonary rehabilitation in Hospital Pulido Valente in Lisbon, Portugal, were invited to participate in the study by CDS between January 2017 and October 2019. Study design was cross-sectional with voluntary convenience sampling (*n* = 100) excluding patients with pleural effusion, infectious disease, unstable cardiac disease, neurologic or musculoskeletal conditions affecting exercise performance and cognitive deficit or psychiatric disease. All patients gave informed consent prior to any proceeding and ethical approval was obtained from the Ethics Committee of Centro Hospitalar Universitário Lisboa Norte, and Centro Académico de Medicina de Lisboa (number 02/17). The trial was registered as NCT03930511 at database clinicaltrials.gov.

The first step was the subjective PADL assessment. Patients were questioned about current physical activity habits (none, or if some on what frequency and modality) and the International Physical Activity Questionnaire [17] (IPAQ) was applied. The IPAQ long form (31 items) assesses PADL across the domains of work, transport, domestic and gardening, and leisure time. Patients answered about frequency and duration of moderate intensity and vigorous intensity activities of at least 10 min. Score considers the metabolic equivalent of task (MET) and is presented as MET.min/week (MET level × minutes of activity × events per week). Guidelines procedure and scoring protocol can be accessed from International Physical Activity Questionnaire [18]. IPAQ categorizes PADL as low (less than 600 MET.min/week), moderate (between 600 and 3000 MET.min/week) and high (at least 3000 MET.min/week). An additional score provided is sitting time (including weekdays and weekends) as an indicator of sedentary behavior.

The second step was the objective PADL assessment. We applied a previously validated method as mentioned [16], and patients were given an Android smartphone with an incorporated accelerometer (Vodafone Smart 4 Turbo, Huawei, China) and also a portable oximeter (Avant 4000™, Nonin Medical, Plymouth, MN, USA) connected by Bluetooth, as illustrated in Figure 1.

This technology supported by Fundação Vodafone Portugal was previously developed for the TELEMOLD project [16], and has now been upgraded to SMARTREAB providing objective PADL assessment of chronic respiratory patients. The method for using the triaxial accelerometer data to calculate energy expenditure by METs uses the coefficient of variation (CV) of counts of movement per 10 s over a 1-min period at patient’s waist level. It applies a 2-regression model with an exponential curve for activities like walking or running where CV ≤ 10, and a cubic curve for everyday activities where CV > 10, as fully described by Crouter [19]. Daily activities have a more erratic movement pattern, resulting in greater variability in counts over time, with the model also adjusted by a linear interpolation for activities between 1 and 2.4 METs.

Patients were instructed by a physiotherapist about telemonitoring equipment usage for a 4-day [4] period including weekdays, weekend and excluding night sleeping hours, and to register their daily activities in a diary. Finger or ear oximeter interface was provided according to patient preference. Heart rate, transcutaneous oxygen saturation (SpO_2_) and MET were monitored, with patient’s access to instant data on the smartphone screen.

SMARTREAB software application presents data per each category as a percentage of total given the telemonitoring timeframe, absolute instant values and allows recording qualitative notes. Figure 2 presents an example of SMARTREAB physical activity telemonitoring, as received remotely at the hospital. 

Data collected were processed and sent by GSM/3G/4G to the hospital database server with restricted accessibility by security password associated to different user privileges. Telemonitoring data could be accessed during real-time assessment and also retrospectively when patients were interviewed to describe activities performed during the 4-day telemonitoring period. Within one to three days after telemonitoring, patients would come to the hospital to crossmatch their written diary with the transcutaneous oxygen desaturation periods (SpO_2_ < 90%) manually identified by the physiotherapist. Details about each oxygen desaturation period were recorded in SMARTREAB with patient´s information about: type of activity, intensity, duration, posture, respiratory symptoms and perceived exertion. Data were analyzed and interpreted with patient active involvement in major conclusions, leading to a team definition of goals and design of a pulmonary rehabilitation program.

Patients were asked about socio-demographic characteristics and clinical information was collected from patient hospital clinical record (this included smoking status, patient comorbidities and prescribed long-term oxygen therapy or nocturnal non-invasive ventilation). 

Statistical analysis and data management were performed using the Statistical Package for the Social Sciences (SPSS) version 25.0 (SPSS Inc., Chicago, IL, USA). A formal sample size was not calculated since this was a convenience sample study of 100 chronic respiratory patients. Descriptive statistics of frequencies were presented as percentages, and quantitative variables were expressed as mean, standard deviation, median and inter-quartile ranges. Associations between categorical variables were analyzed using chi-squared tests or Fisher exact tests (for iPAQ category with smoking status and with home accessibility, due to categories with low numbers), and association between continuous variables was analyzed with Spearman’s correlation coefficient (*ρ*). A *p* value of less than 0.05 was considered statistically significant.

## 3. Results

### 3.1. Patients’ Characteristics 

Recruitment included 127 people, with 22 patients declining and 5 patients discontinued from the study, as described in detail in Figure 3.

One hundred participants were all admitted to initiate pulmonary rehabilitation, had a mean age of 66.1 ± 9.8 years and 50% were male. Most prevalent chronic respiratory conditions were COPD (41%) and ILD (22%, mainly idiopathic pulmonary fibrosis), followed by asthma (15%), bronchiectasis (10%) and others (12%, including post-thoracic surgery, lung cancer, tuberculosis sequelae, lung disorder associated with connective tissue disease and pulmonary ossification). 

Most patients were married (67%), had adult children (80%), had grandchildren (61%) and relied on family support (95%). The majority were high-school graduates (55%) and were already retired (70%). Patients’ most frequent living environment was a flat street (62%) with stairs accessibility (57%) to an apartment (82%). 

Patients reported a median of 5 comorbidities, and the 3 most frequent were: arterial hypertension (56%), dyslipidemia (40%) and anxiety or depression (29%). Along with patient major chronic respiratory disease, the most frequent respiratory comorbidities were: bronchiectasis (28%), chronic respiratory failure (26%) and pulmonary fibrosis (20%). Sixty percent of the patients had a history of smoking and 8% were active smokers. Patients reported a mean of 6 h of sleep per night (13% of patients considered themselves to have bad sleeping quality), 29% of patients on long-term oxygen therapy and 12% on nocturnal non-invasive ventilation. 

Patients’ characteristics per chronic respiratory condition are presented on Table 1. 

### 3.2. Physical Activity Subjective Assessment

Only 40% of patients reported that they are currently engaging in physical activity, mainly walking around the block or a park. No current physical activity was reported by 50% of men and 70% of women. 

PADL self-reported by IPAQ ranged from 0 to 15,160 MET.min/week, with a mean amount of 2844 ± 2925 MET.min/week and a median (interquartile range) of 2043 (876–3684) MET.min/week. According to IPAQ, 20% of patients had low PADL (less than 600 MET.min/week), 49% of patients had moderate PADL (between 600 and 3000 MET.min/week) and 31% of patients had high PADL (more than 3000 MET.min/week). Subjective PADL assessment per chronic respiratory disease is presented on Table 2.

When PADL was analyzed in terms of IPAQ category (low, moderate or high physical activity levels), 50% of patients with COPD, ILD and asthma reported a moderate level of PADL, while 50% of patients with bronchiectasis and others reported high levels of PADL. There was an association with smoking status (*p* = 0.02), in which active smokers demonstrated less activity than non-smokers, and an association with long-term oxygen therapy (*p* = 0.005), in which patients who use oxygen had lower levels of physical activity. There was no significant association between IPAQ categories and home accessibility (*p* = 0.129), in terms of having an elevator, stairs or no barriers accessing home; and there was no association with the total number of comorbidities (*p* = 0.109). Patients with low or moderate PADL did not report vigorous intensity activity, and those with low PADL were mainly active with regard to transportation, including walking time. On the other hand, patients with high PADL were active workers, dedicating more time to the household, gardening, leisure and sports, including some vigorous intensity activities time. 

When PADL was analyzed in terms of IPAQ score (MET.min/week), the COPD group had the lowest score, and more than 75% of patients in this group reported physical activity levels below 3000 MET.min/week. Regarding patients with bronchiectasis, 25% had less than 600 MET.min/week and more than 25% surpassed 6000 MET.min/week, which means there is a wide range of reported physical activity in this group. The associations between IPAQ total score and its component subscores are presented as Appendix A. Global reported physical activity energy expenditure was positively associated with moderate activity (*ρ* = 0.714, *p* < 0.0005), and this level of activity was strongly correlated with reported gardening and domestic activity energy expenditure (*ρ* = 0.879, *p* < 0.0005). 

Self-reported sedentary time ranged from 30 min to 15 h per day, with a mean sedentary time of 5.8 ± 2.7 h/day. COPD and ILD patients reported a higher mean daily sitting time, compared to patients with asthma and bronchiectasis.

### 3.3. Physical Activity Objective Assessment

Patients had a mean of 33 ± 12 h of PADL telemonitoring on waking hours during a 4-day period, ranging between 11 to 61 h of synchronous recorded assessment. Each patient had less than 15% of telemonitoring time with invalid records, mainly related to lack of battery, either of the smartphone or the oximeter, and corrected by an alert from the physiotherapist. Percentage of SMARTREAB telemonitoring time is presented on Table 3, discriminating PADL intensity by chronic respiratory condition, MET category and SpO_2_ below 90% (oxygen desaturation period). Maximum MET ranged from 1.51 to 4.64 METs, with mean values per chronic respiratory disease presented on Table 3. 

Results show that patients with COPD, ILD, asthma and bronchiectasis spent about 80% of their daytime on activities below 2 METs, including the IPAQ reported sitting time. Moreover, 99.6% of their daytime was spent on activities below 3 METs. The associations between physical activity telemonitoring results with total of comorbidities or respiratory comorbidities are presented as Appendix A. There was a large variability in level of physical activity among people with different number of comorbidities. 

Furthermore, SMARTREAB showed 93% of patients with daily oxygen desaturation episodes (SpO_2_ < 90%), a finding unobserved by usual clinical setting assessment. When admitted to the pulmonary rehabilitation program, patients with ILD presented more time of daily desaturation episodes, followed by COPD, asthma and bronchiectasis patients, as presented on Table 3.

### 3.4. Combined Analysis of Subjective and Objective Physical Activity Assessment 

The evidence encountered was that whatever the IPAQ score, category or reported physical activity habits, objective assessment according to SMARTREAB showed patients spent at least a mean of 70% of the daytime with activities below 2 METs, as shown on Table 4 with a stratified analysis by IPAQ category. 

The associations between percentage of telemonitoring time spent on activities below 2 METs, between 2 and 3 METs and above 3 METs with IPAQ subscores is presented as Appendix A. There were no relevant associations. Results for combined subjective and objective PADL assessments per chronic respiratory disease are presented in Figure 4 and more detailed data as Appendix A.

According to our telemonitoring results, all self-reported IPAQ categories (low, moderate, high) had 50% of COPD, ILD, asthma and bronchiectasis patients spending 80% of time on activities below 2 METs, about 20% of time on activities between 2 and 3 METs, and less than 1% of time on activities above 3 METs. 

About 75% of patients with COPD, ILD and asthma reported a high level of PADL but had more than 50% of telemonitoring time on activities below 2 METs. Moreover, at least 25% of patients with COPD, ILD, asthma and bronchiectasis had self-reported a moderate or high level of PADL and were found to have almost 100% telemonitoring time on activities below 2 METs. 

More than 50% telemonitoring time spent on activities between 2 and 3 METs was found on 25% of patients self-reporting a high level of PADL. Most bronchiectasis patients had less than 35% telemonitoring time spent on activities between 2 and 3 METs, whatever IPAQ category. Comparing self-reported moderate to high PADL on time spent on activities between 2 and 3 METs, asthma and ILD groups had an increase, which did not happen with the COPD and bronchiectasis groups. 

Less than 1% telemonitoring time was spent on activities above 3METs for all self-reported PADL categories, except for the other chronic respiratory disease group, with 50% of self-reported high PADL patients with 0.5% to 2% telemonitoring time spent on activities above 3METs. 

### 3.5. Results Applied to Designing Patient-Tailored Pulmonary Rehabilitation Programs 

SMARTREAB was applied as an innovative telemedicine solution with significant patient involvement in telemonitoring data analysis and interpretation, contributing to single-case major conclusions and personal rehabilitation goals commitment. This means that such a process was unique for each patient and could be applied to design individually-tailored pulmonary rehabilitation.

Figure 5 represents SMARTREAB rationale with the example of two clinical cases: a female patient with COPD that only desaturated during the final 10 min of 2 h gardening, and the case of a male patient with bronchiectasis who had desaturation periods during family support routines.

## 4. Discussion

The main findings of the current study were that patients with chronic respiratory disease engage on daily activities that only raise up to 3-fold their basal metabolism (3 METs), spending at least 70% daytime on activities below 2 METs, including 6 h on sitting position. Moreover, transcutaneous oxygen desaturation occurred in 93% of patients during the 4-day measuring period.

When questioned about physical activity habits, 6 in each 10 chronic respiratory patients declared to be inactive, being 50% men and 70% women. This is in accordance with national statistics of physical inactivity of 55% for men and 72% for women [20], and also with results reported by patients with COPD [21], ILD [11], asthma [22] and bronchiectasis [23]. 

When considering PADL as a patient-reported outcome assessed by IPAQ, our results showed that chronic respiratory patients with higher levels of PADL were active workers, used stairs to enter the home, and dedicated more time to household, gardening, leisure and sports, including some vigorous intensity activity time. On the other hand, patients with chronic respiratory disease who reported low or moderate PADL were solely active on walking time, and these included active smokers, patients on long-term oxygen therapy and with several comorbidities. 

These patient characteristics are aligned with an interesting published framework analyzing physical capacity and physical activity as a quadrant concept which identified: 34% of patients that “can’t do, don’t do”, 31% of patients that “can do, do do”, 21% of patients that “can’t do, do do” and 14% of patients that “can do, don’t do” [24]. Such different PADL profiles, if assessed as a baseline general physical condition vital sign, provide important input for specific strategies on exercise counselling and chronic disease management [2] that might increase pulmonary rehabilitation efficiency as a process.

Regardless of physical activity time, we found chronic respiratory patients reporting a daily average of 6 h on a sitting position. Such a health problem should not be neglected in clinical practice, as recent research indicates an increased risk of cardiovascular mortality for more than 6 h/day sitting time, and also an increased risk for all-cause mortality for more than 3–4 h/day of television viewing time [25]. Even adjusting sedentary behavior for moderate-to-vigorous physical activity, there is evidence in COPD patients that sedentary behavior is an independent predictor of mortality [26]. Hence, decreasing sedentary time should be an individually predefined goal on pulmonary rehabilitation, with patient awareness and commitment to success, as we came to apply on our current clinical practice.

Considering PADL as an objective outcome using SMARTREAB, chronic respiratory patients engage on daily activities that only raise up to 3-fold their basal metabolism (3 METs), and spend a minimum of 70% daytime on activities below 2 METs. Such decreased physical activity was consistent among all chronic respiratory conditions and it should not be disregarded, as it may cover a great variability on patients’ daily physical activity patterns. This has been elegantly demonstrated by Mesquita et al. with the identification of 5 main COPD clusters on such matters: couch potatoes, high sedentary, sedentary movers, sedentary exercisers and busy bees [5]. 

A combined analysis of IPAQ and SMARTREAB revealed that all self-reported PADL categories had 50% of COPD, ILD, asthma and bronchiectasis patients spending 80% of their time on activities below 2 METs, about 20% of time on activities between 2 and 3 METs, and less than 1% of time on activities above 3 METs. This distribution of PADL time patterns should be acknowledged as a personal treatable trait when designing the individually-tailored physical activity intervention as a model of care [27].

For clinical purposes, each chronic respiratory patient was interviewed to describe activities performed during telemonitoring. This combination of data collected by SMARTREAB and qualitative patient input was essential to clarify the oxygen desaturation periods shown. Such methodology opportunity is crucial for patients’ security, as SMARTREAB showed 93% of patients with daily desaturation episodes. This finding was unobserved by usual clinical setting assessment, revealing a pressing need and interest for continuous research.

Considering all this, each of the 100 patients was involved in analyzing and interpreting telemonitoring data, contributing to single-case major conclusions, and committing to individual goals aligned with the design of pulmonary rehabilitation intervention. Patient involvement and benefit was achieved, with high satisfaction and impact on daily life, successfully accomplishing the pulmonary rehabilitation purpose. 

Overall, this study presents evidence and enhances the importance of appropriate PADL objective measures, specific to behaviors in free-living environments as recommended by the most recent systematic review of reviews [28]. We found that telemonitoring PADL with SMARTREAB provided important information about a general physical condition as a vital sign to consider when designing a patient-tailored pulmonary rehabilitation intervention. Such telemedicine solution combined with patient engagement enhances interested attendance, treatment compliance, and long-term program adherence, fostering synergies on physical activity and pulmonary rehabilitation agendas [29].

## 5. Conclusions

Given the importance of physical inactivity as a consequence of chronic respiratory disease and a potential modifiable risk factor, it is recommended to promptly address this outcome as a vital sign of patients’ general physical condition in order to optimize an exercise-based intervention. Physical activity telemonitoring revealed that chronic respiratory patients engage on daily activities raising up to 3-fold of basal metabolism, regardless of the IPAQ scoring and category as a patient subjective reported outcome. SMARTREAB was demonstrated to be an innovative methodology to measure PADL as a vital sign combining oximetry, accelerometry and qualitative patient data, providing important input for designing patient-tailored pulmonary rehabilitation. 

## Figures and Tables

**Figure 1 jcm-09-02450-f001:**
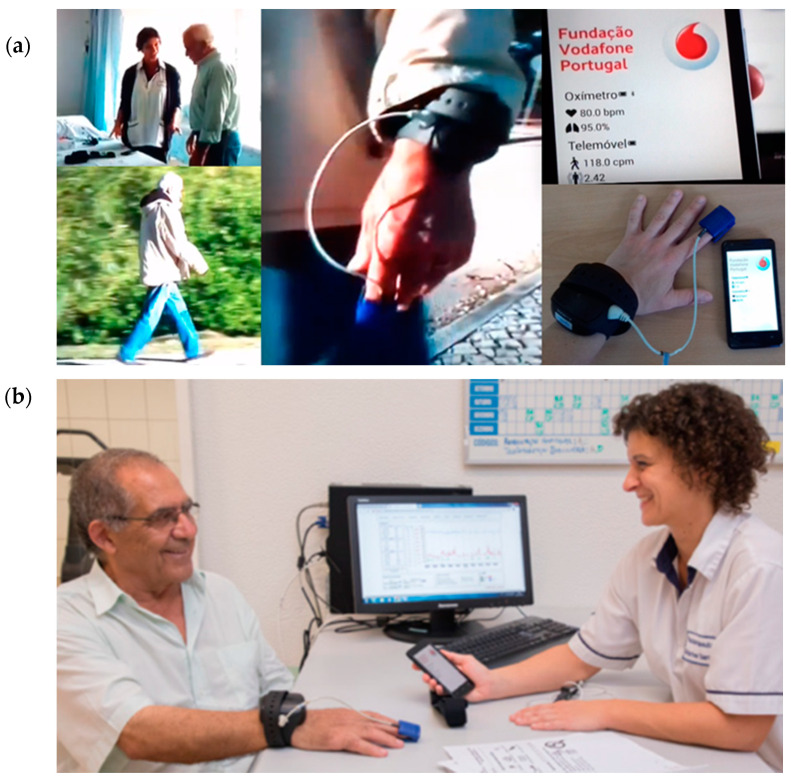
(**a**) Patient receiving instructions on how to use the SMARTREAB equipment: an Android smartphone with an accelerometer (to be used at waist level using a belt with an inside pocket), connected by Bluetooth to a portable oximeter with finger or ear sensor. During telemonitoring patients have access to the smartphone screen information indicating % of transcutaneous oxygen saturation, bpm of heart rate and metabolic equivalent of task (MET) of physical activity. (**b**) Together patient and physiotherapist analyze retrospective telemonitoring data integrating important input for the design of the patient-tailored pulmonary rehabilitation. (The patients and healthcare professional shown consented to be photographed and such images to be used for research publication purposes).

**Figure 2 jcm-09-02450-f002:**
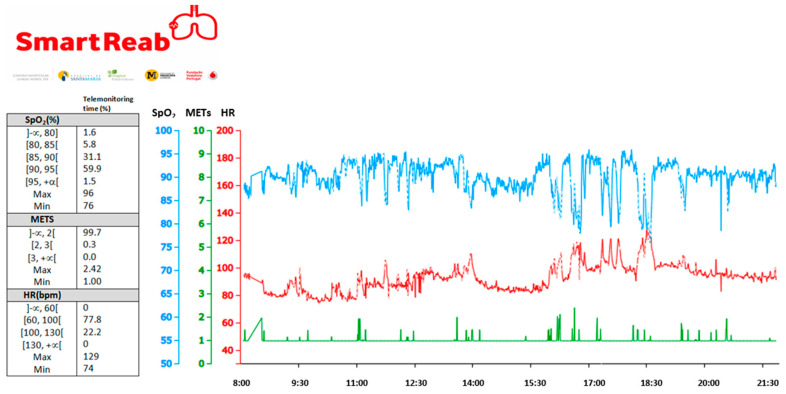
Example of SMARTREAB telemonitoring transcutaneous oxygen saturation (SpO_2_), heart rate (HR) and physical activity intensity (MET) on a single daytime period of a chronic respiratory patient. This example shows 38.5% of telemonitoring time with oxygen desaturation with daily physical activity below 2.42 METs. The increased heart rate and decreased SpO_2_ between 15:30 and 18:30 corresponds to performing household chores such as vacuuming, mopping, dusting and tidying up.

**Figure 3 jcm-09-02450-f003:**
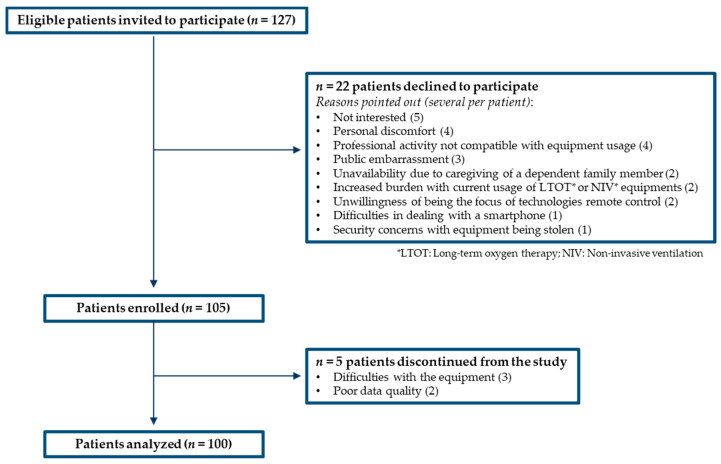
Enrollment flow diagram.

**Figure 4 jcm-09-02450-f004:**
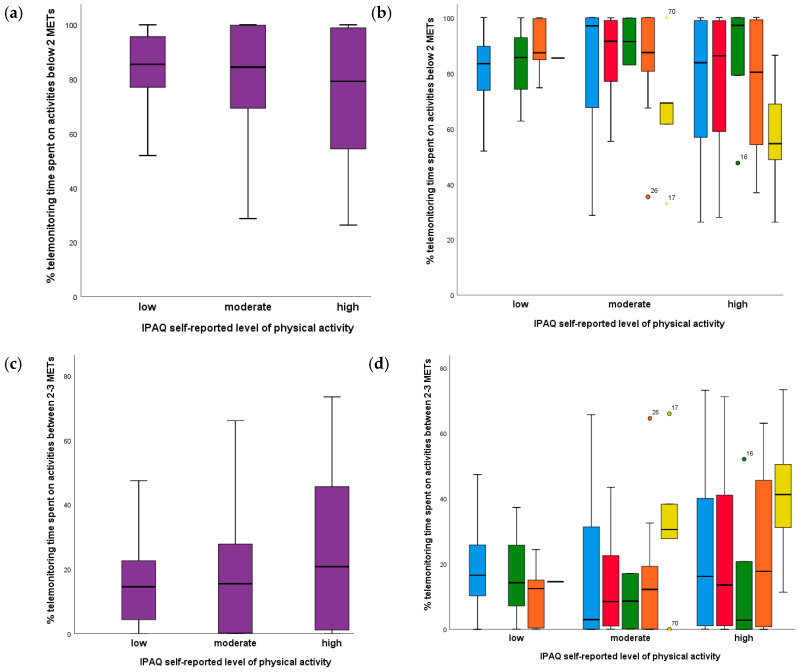
Telemonitoring time: (**a**) spent on activities below 2 METs by self-reported IPAQ level of physical activity; (**b**) spent on activities below 2 METs by self-reported IPAQ level of physical activity per chronic respiratory disease; (**c**) spent on activities between 2 and 3 METs by self-reported IPAQ level of physical activity; (**d**) spent on activities between 2 and 3 METs by self-reported IPAQ level of physical activity per chronic respiratory disease; (**e**) spent on activities above 3 METs by self-reported IPAQ level of physical activity; (**f**) spent on activities above 3 METs by self-reported IPAQ level of physical activity per chronic respiratory disease.

**Figure 5 jcm-09-02450-f005:**
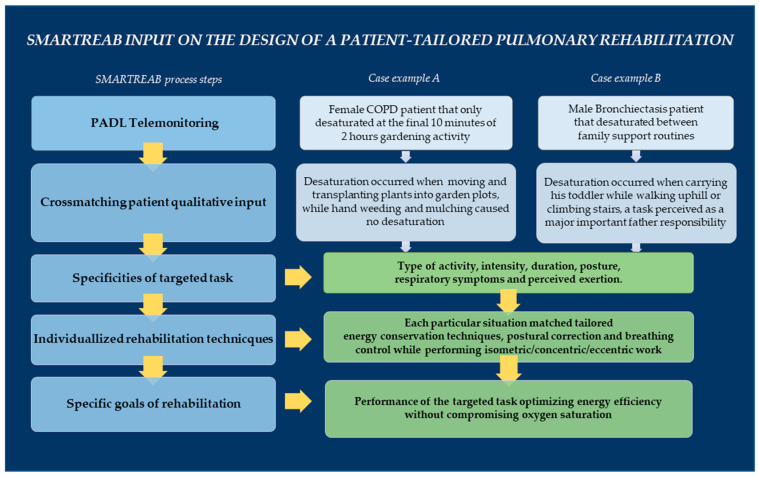
SMARTREAB rationale to operationalize single-patient PADL telemonitoring data as an input to set individually-tailored pulmonary rehabilitation goals. All patients, including cases A and B described, were the center of identification and characterization of targeted desaturation tasks, and thereby actively involved on specific goal-setting and commitment to intervention success.

**Table 1 jcm-09-02450-t001:** Patients’ characteristics per chronic respiratory disease.

	Total	COPD	ILD	Asthma	Bronch.	Others
	*n* = 100	*n* = 41	*n* = 22	*n* = 15	*n* = 10	*n* = 12
	%	*n* (%)	*n* (%)	*n* (%)	*n* (%)	*n* (%)
**Male gender**	50	31 (75.6)	8 (36.4)	4 (26.7)	3 (30.0)	4 (33.3)
**Age** (years)						
mean	66.1	68	63.4	66	68	63.2
*±*SD	±9.8	±7.2	±9.5	±11.9	±13.2	±11.9
**Social Status**						
Single	10	4 (9.8)	4 (18.2)	1 (6.7)	0 (0.0)	1 (8.3)
Married	67	27 (65.9)	14 (63.6)	10 (66.7)	9 (90.0)	7 (58.3)
Divorced	10	6 (14.6)	2 (9.1)	0 (0.0)	0 (0.0)	2 (16.7)
Widow	13	4 (9.8)	2 (9.1)	4 (26.7)	1 (10.0)	2 (16.7)
**Children**	80	31 (75.6)	19 (86.4)	14 (93.3)	8 (80.0)	8 (66.7)
**Grandchildren**	61	25 (61.0)	14 (63.3)	10 (66.7)	6 (60.0)	6 (50.0)
**Education level**						
Absent to school	4	1 (2.4)	0 (0.0)	0 (0.0)	2 (20.0)	1 (8.3)
Primary school level	23	8 (19.5)	5 (22.7)	6 (40.0)	2 (20.0)	2 (16.7)
Secondary school level	55	23 (56.1)	12 (54.5)	9 (60.0)	5 (50.0)	6 (50.0)
High degree level	18	9 (22.0)	5 (22.7)	0 (0.0)	1 (10.0)	3 (25.0)
**Employment status**						
Active	19	5 (12.2)	4 (18.2)	4 (26.7)	3 (30.0)	3 (25.0)
Sick leave	10	3 (7.3)	4 (18.2)	2 (13.3)	0 (0.0)	1 (8.3)
Retired	70	32 (78.0)	14 (63.6)	9 (60.0)	7 (70.0)	8 (66.7)
Unemployed	1	1 (2.4)	0 (0.0)	0 (0.0)	0 (0.0)	0 (0.0)
**Social support**						
Family	95	38 (92.7)	21 (95.5)	15 (100)	9 (90.0)	12 (100)
Home-assistance	3	2 (4.9)	1 (4.5)	0 (0.0)	0 (0.0)	0 (0.0)
None	2	1 (2.4)	0 (0.0)	0 (0.0)	1 (10.0)	0 (0.0)
**Living environment**						
House	18	7 (17.1)	4 (18.2)	1 (6.7)	5 (50.0)	1 (8.3)
Flat	82	34 (82.9)	18 (81.8)	14 (93.3)	5 (50.0)	11 (91.7)
**Home accessibility**						
Stairs	57	20 (48.8)	12 (54.5)	9 (60.0)	10 (100)	6 (50.0)
Elevator	39	19 (46.3)	8 (36.4)	6 (40.0)	0 (0.0)	6 (50.0)
No physical barriers	4	2 (4.9)	2 (9.1)	0 (0.0)	0 (0.0)	0 (0.0)
**Living street**						
Flat area	62	26 (63.4)	14 (63.6)	8 (53.3)	8 (80.0)	6 (50.0)
With slope	38	15 (36.6)	8 (36.4)	7 (46.7)	2 (20.0)	6 (50.0)
**Smoking Status**						
Non smoker	40	2 (4.9)	11 (50.0)	9 (60.0)	10 (100)	8 (66.7)
Former smoker	52	34 (82.9)	10 (45.5)	4 (26.7)	0 (0.0)	4 (33.3)
Active smoker	8	5 (12.2)	1 (4.5)	2 (13.3)	0 (0.0)	0 (0.0)
**Number of comorbidities**						
median (IQR)	5 (4)	5 (3)	5 (3)	4 (5)	5 (5)	6 (7)
**LTOT**	29	16 (39.0)	11 (50.0)	0 (0.0)	1 (10.0)	1 (8.3)
**Nocturnal NIV**	12	5 (12.2)	3 (13.6)	3 (20.0)	1 (10.0)	0 (0.0)

COPD: Chronic Obstructive Pulmonary Disease; ILD: Interstitial Lung Disease; Bronch.: Bronchiectasis; Others: post-thoracic surgery, Lung Cancer, Tuberculosis sequelae, Lung Disorder associated with Connective Tissue Disease and Pulmonary Ossification; SD: standard deviation; IQR: interquartile range; LTOT: Long-term Oxygen Therapy; NIV: Non-Invasive Ventilation.

**Table 2 jcm-09-02450-t002:** Subjective International Physical Activity Questionnaire (IPAQ) physical activity in daily life (PADL) assessment per chronic respiratory disease.

	Total	COPD	ILD	Asthma	Bronch.	Others
	*n* = 100	*n* = 41	*n* = 22	*n* = 15	*n* = 10	*n* = 12
**IPAQ category** (*n*,%)						
Low	20 (20.0)	11 (26.8)	5 (22.7)	0 (0.0)	3 (30.0)	1 (8.3)
Moderate	49 (49.0)	22 (53.7)	12 (54.5)	8 (53.3)	2 (20.0)	5 (41.7)
High	31 (31.0)	8 (19.5)	5 (22.7)	7 (46.7)	5 (50.0)	6 (50.0)
**IPAQ score**						
(MET.min/week)						
Mean	2844	2318	2602	3927	3340	3320
±SD	±2925	±2667	±3470	±2596	±3713	±2253
P 25%	876	620	908	1872	420	1413
P 50%	2043	1900	1446	2523	2000	3214
P 75%	3684	2744	3036	6642	6480	4922
**IPAQ sitting time**	5.8 ± 2.7	6.4 ± 2.8	6.1 ± 2.7	4.3 ± 2.3	5.2 ± 1.8	5.8 ± 3.1
(h/day; mean ± SD)						

PADL: physical activity in daily life; IPAQ: International Physical Activity Questionnaire; SD: standard deviation; P25%: first quartile; P50%: second quartile (median); P75%: third quartile; COPD: Chronic Obstructive Pulmonary Disease; ILD: Interstitial Lung Disease; Bronch.: Bronchiectasis; Others: post-thoracic surgery, Lung Cancer, Tuberculosis sequelae, Lung Disorder associated with Connective Tissue Disease and Pulmonary Ossification.

**Table 3 jcm-09-02450-t003:** Objective SMARTREAB PADL assessment per chronic respiratory disease.

	Total	COPD	ILD	Asthma	Bronch.	Others
	*n* = 100	*n* = 41	*n* = 22	*n* = 15	*n* = 10	*n* = 12
**PADL Maximum MET**						
Mean	2.9	2.85	2.85	2.89	2.6	3.48
±SD	±0.68	±0.65	±0.74	±0.45	±0.61	±0.77
**% telemonitoring time**						
PADL < 2 METs						
Mean	80	81.4	83.3	81.3	85.5	63.2
±SD	±21.0	±20.4	±19.5	±22.1	±18.2	±21.6
PADL from 2 to 3 METs						
Mean	19.6	18.1	16.5	18.5	14.5	35.5
±SD	±20.5	±19.6	±19.5	±21.9	±18.1	±21.5
PADL > 3 METs						
Mean	0.4	0.5	0.2	0.2	0	1.3
±SD	±1.2	±1.3	±0.4	±0.3	±0.1	±2.4
**% telemonitoring time**						
SpO_2_ < 90%						
Mean	13.6	13.9	20.4	7.3	4.9	15.2
±SD	±19.1	±19.0	±20.9	±7.6	±6.0	±28.7

PADL: physical activity in daily life; COPD: Chronic Obstructive Pulmonary Disease; ILD: Interstitial Lung Disease; Bronch.: Bronchiectasis; Others: Post-thoracic Surgery, Lung Cancer, Tuberculosis sequelae, Lung Disorder associated with Connective Tissue Disease and Pulmonary Ossification; SD: standard deviation.

**Table 4 jcm-09-02450-t004:** Subjective IPAQ and objective SMARTREAB PADL assessments.

	% Telemonitoring Time (Mean ± SD) with SMARTREAB		
	<2 METs	from 2 to 3 METs	≥3 METs
**IPAQ category**			
Low (*n* = 20)	83.2 ± 14.5	16.6 ± 14.4	0.2 ± 0.3
Moderate (*n* = 49)	83.0 ± 19.2	16.6 ± 18.7	0.4 ± 1.0
High (*n* = 31)	73.2 ± 25.7	26.1 ± 25.2	0.7 ± 1.8

PADL: physical activity in daily life; MET: metabolic equivalence of task; IPAQ: International Physical Activity Questionnaire; SD: standard deviation.

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
