# Peer review of "Novel Input for Designing Patient-Tailored Pulmonary Rehabilitation: Telemonitoring Physical Activity as a Vital Sign—SMARTREAB Study"

_jcm, 2020, doi:10.3390/jcm9082450_

Round 1

Reviewer 1 Report

The authors present the results of a study to compare daily physical activity performed by patients with chronic respiratory disorders, presenting a system for the comprehensive measurement of physical activity considering oximetry, accelerometry, and qualitative patient data.

The manuscript is well structured and written. The introduction presents in a synthetic and adequate way the background and motivation of the work, focusing the study in the context of a previous project (TELEMOLD) whose development laid the foundations for the experiments and results presented.

The assessment of physical exercise in patients with respiratory diseases, either as an element for the adjustment of therapy (rehabilitation, home oxygen therapy, etc.) or as a predictive element of episodes of exacerbation of the disease, is a recurrent topic of interest for the community of researchers in the respiratory field.

The results presented are interesting and may help in a more adequate prescription of therapy or in the design of personalized action plans for patients. However, after reviewing the manuscript, some doubts arise, mainly of a methodological nature, which I would like to see clarified by the authors.

The authors state in section 2 that the patient used the accelerometer of the Android mobile device to estimate the metabolic equivalent of the tasks. This implies that the patient carried the mobile phone with him/her during the whole recording period. In contrast, in the TELEMOLD study, a specific accelerometer device was used, placed on the waist. Does this change affect the results obtained? Where was the mobile phone placed for each patient? Was a common location established? What algorithm/method was used to calculate the METs from the accelerometer signal of the mobile phone?

Regarding oximetry, the problem of this technique to record in motion is well known, given the high amount of artifacts present in the measurement, even more so for recording in high light environments. In the TELEMOLD study, a percentage of the measurements taken could not be used for these reasons and even complete records of some patients had to be discarded. These aspects are not detailed in the paper, neither in the methodology nor in the results section. What percentage of the Spo2 data recorded could not be analyzed? Did any patient have to be discarded for this reason?

Finally, the heart rate, recorded in the experiments, is not analyzed in the study. Didn't it provide any relevant information?

In my opinion, the table provided as supplementary material should be included in the manuscript.

Apart from these considerations, which should be clarified, the findings presented in the discussion and reinforced in the conclusions are interesting because of the clinical implications they may entail.

Author Response

Response to Reviewer 1 Comments

The authors present the results of a study to compare daily physical activity performed by patients with chronic respiratory disorders, presenting a system for the comprehensive measurement of physical activity considering oximetry, accelerometry, and qualitative patient data.

The manuscript is well structured and written. The introduction presents in a synthetic and adequate way the background and motivation of the work, focusing the study in the context of a previous project (TELEMOLD) whose development laid the foundations for the experiments and results presented.

The assessment of physical exercise in patients with respiratory diseases, either as an element for the adjustment of therapy (rehabilitation, home oxygen therapy, etc.) or as a predictive element of episodes of exacerbation of the disease, is a recurrent topic of interest for the community of researchers in the respiratory field.

The results presented are interesting and may help in a more adequate prescription of therapy or in the design of personalized action plans for patients. However, after reviewing the manuscript, some doubts arise, mainly of a methodological nature, which I would like to see clarified by the authors.

Point 1: The authors state in section 2 that the patient used the accelerometer of the Android mobile device to estimate the metabolic equivalent of the tasks. This implies that the patient carried the mobile phone with him/her during the whole recording period. In contrast, in the TELEMOLD study, a specific accelerometer device was used, placed on the waist. Does this change affect the results obtained?

Response 1: In the TELEMOLD study patients carried at waist level both the mobile phone and the external accelerometer connected by Bluetooth. In the SMARTREAB study devices were upgraded to a hardware solution with the accelerometer incorporated in the Android smartphone, also carried by patients at waist level. Both studies had the same biomechanical anatomic reference for counts per minute by accelerometery, and patients were given instructions to wear the equipment at waist level at both studies, using a belt with an inside pocket. Our technical team also proceeded with preliminary tests comparing results between TELEMOLD and SMARTREAB equipment scenarios, and such change did not affect the results obtained.

Point 2: Where was the mobile phone placed for each patient?

Response 2: The Android smartphone was used at waist level using a belt with an inside pocket, as described in Figure 1 caption in the Experimental section, line 113.

Point 3: Was a common location established?

Response 3: All patients were instructed to use the Android smartphone at waist level using a belt with an inside pocket. Patients were also explained that such anatomical reference was important to keep with during daily activities because of the accuracy of measuring counts per movement to determine physical activity intensity.

Point 4: What algorithm/method was used to calculate the METs from the accelerometer signal of the mobile phone?

Response 4: The authors added to the Experimental section lines 123-130 clarifying readers of such matter: “The method for using the triaxial accelerometer data to calculate energy expenditure by METS uses the coefficient of variation (CV) of counts of movement per 10 seconds over a 1-minute period at patient’s waist level. It applies a 2-regression model with an exponential curve for activities like walking or running where CV ≤ 10, and a cubic curve for everyday activities where CV > 10, as fully described by Crouter [18]. Daily activities have a more erratic movement pattern, resulting in greater variability in counts over time, with the model also adjusted by a linear interpolation for activities between 1 and 2.4 METS.” The mentioned algorithms are fully described at Crouter SE, Clowers KG, Bassett DR Jr. A novel method for using accelerometer data to predict energy expenditure. J Appl Physiol 2006, 100 (4), 1324-1331. This last reference was also added as number 18 of list of references.

Point 5: Regarding oximetry, the problem of this technique to record in motion is well known, given the high amount of artifacts present in the measurement, even more so for recording in high light environments. In the TELEMOLD study, a percentage of the measurements taken could not be used for these reasons and even complete records of some patients had to be discarded. These aspects are not detailed in the paper, neither in the methodology nor in the results section. What percentage of the Spo2 data recorded could not be analyzed?

Response 5: Authors have added to the Results section lines 243-245: “Each patient had less than 15% of telemonitoring time with invalid records, mainly related to lack of battery, either of the smartphone or the oximeter, and corrected by an alert from the physiotherapist.”.

Point 6: Did any patient have to be discarded for this reason?

Response 6: Authors have added to the Experimental section lines 88-92: “Recruitment included 127 people, with 22 patients declining to participate due to personal discomfort, public embarrassment or difficulties in dealing with a smartphone and remote technologies. There were 5 patients who consented to participate but were discontinued from the study given practical difficulties to address the equipment and related poor quality of the telemonitoring output data.”.

Point 7: Finally, the heart rate, recorded in the experiments, is not analyzed in the study. Didn't it provide any relevant information?

Response 7: Patient telemonitored heart rate provided relevant information that authors will develop on a succeeding manuscript comparing results of oxygen desaturation and heart rate on physical activity of daily life with heart rate monitored at hospital six-minute walk test.

Point 8: In my opinion, the table provided as supplementary material should be included in the manuscript.

Response 8: Thank for this suggestion, these results are indeed important. The authors decided to present these results in the manuscript in the Results section with Figure 3 on line 282, but kept the detailed information of Table S1 as supplementary online material.

Apart from these considerations, which should be clarified, the findings presented in the discussion and reinforced in the conclusions are interesting because of the clinical implications they may entail.

Reviewer 2 Report

Santos and colleagues present a descriptive observational study of telemonitoring of daily physical activity in patients with chronic respiratory disease. They use telemonitoring to quantify heart rate, activity, and oxygen saturation over a 4 day monitoring period. They compare this to a validated questionnaire (IPAQ), and determine correlations between the subjective and objective methods. Finally, they describe how the telemonitoring data can be used to create an individualised pulmonary rehabilitation plan.

The study presents an interesting way of integrating objective activity/oxygen monitoring into pulmonary rehabilitation. However, this most interesting part gets lost in a lot of detail and multiple comparisons/correlations. The purpose of the study therefore is a bit unclear. Is this about validating telemonitoring compared to subjective questionnaires? Is this about the pulmonary rehab planning? Due to this lack of clear aims, the ‘novel’ aspects of this study are not really clear to the reader and I think a lot of work needs to be done on this manuscript before it is ready for publication.

General comments:

  • Introduction – there is a lot of detail about findings in each disease category. This level of detail is not necessary to describe the rationale for this study, and all of this information could be summarised in a single paragraph. The authors should also make the point here that “average activity level” may not be the most appropriate measurement since there is great variability in the types of activities patients do throughout a day - this is a point the authors later make in the Discussion, and I think is the main rationale for real-time activity/oxygen monitoring
  • Analysis – there are a lot of correlations made between variables, including categorical variables. While Pearson correlation is appropriate for normally-distributed continuous variables, I am concerned about the use of Spearman correlation for variable that are categorical/ordinal since there will be many “ties” and Spearman rho does not handle this well. We should be able to see the data plotted rather than just r/rho. Furthermore, if the aim was to validate the SMARTREAB monitoring device against the IPAQ questionnaire, there are probably more appropriate methods than multiple pairwise correlation testing. I suggest formal statistical review either through the Journal, or through the authors’ own institution.
  • Since the most interesting and novel part of this study is how the telemonitoring information is used to direct the rehabilitation goals, I think this should be given more prominence in the Results rather than just being mentioned in the Discussion. The case study given is excellent. It would be great to see a second or even third case study – these could be presented as a “box” in the paper (i.e. a separate panel of text, treated like a figure rather than a table – the Journal may be able to provide examples from recently-published articles).

Specific comments:

  • Abstract, line 28: change “previously” to “prior”
  • Introduction line 48: “advocates stakeholders for action” this doesn’t really make sense
  • Introduction lines 52-70: these 4 paragraphs should be condensed into one, describing the main point that physical inactivity is common in chronic respiratory disease and is associated with poor outcomes
  • Line 56: the number of steps/day in COPD is meaningless without a comparison to healthy controls
  • Line 80: “escalated to SMARTREAB” – this doesn’t really describe how it is different to the TELEMOLD study. Is it truly escalated? Or just extended for a different indication/purpose?
  • Line 86: aim 2) isn’t really an experimental aim. It should be rephrased as a description only
  • Experimental section, line 91: how were the patients selected? Were all patients approached at the time of referral? We should see the number approached, number refused, etc in the results
  • Line 129: briefly describe how MET is calculated from the telemonitoring data
  • Line 143: how did you conduct this interview? Did the patients keep a diary at the time? How long after the monitoring did you interview them?
  • Lines 143-145: the authors should describe how the information was used to develop their rehabilitation goals – who looked at the data, how did they break it down into relevant periods etc (this is where the “box” case studies would be helpful)
  • Results, line 177: Table 1 should include the number of comorbidities. It is also customary to report “n (%)” rather than “% (n)”. “Professional status” is better described as “Employment status”
  • Line 186: “medium” should be “median”
  • Line 193 Table 2: report “n (%)”. Does P25%, P50% etc represent percentiles? If so, please clarify in the table caption.
  • Line 199: “correlation with smoking” what is the smoking variable? Current, pack years, etc?
  • Line 200: “oxygen therapy” is this use of LTOT yes/no?
  • Line 207: “High PADL by IPAQ total score was associated…” What does this mean? What are the variable you are comparing?
  • Line 214: “The previous questionnaire-reported values had a combined analysis with a 4 daytime…” I don’t understand this sentence/analysis.
  • Line 228: “…including the above mentioned sitting time” I don’t see any data on sitting time
  • Line 232: there are a number of mentions of “respiratory comorbidities” but this has not been defined
  • Lines 236-239: What is the purpose of this analysis? I don’t think it is very meaningful
  • Line 240: “…daily oxygen desaturation episodes” how was a desaturation episode defined?
  • Line 250 Table 4: If I understand this correctly, this is a stratified analysis by IPAQ category? If so, this should be stated so the reader understands. The N for each IPAQ category should be shown in the table
  • Discussion lines 279-285: I don’t really see how your data aligns with these patient “clusters”. If the authors believe their data shows something similar, they should explain how/why
  • Lines 298-299: “…as it may be a product of a great variability on patients’ daily physical activity patterns” I don’t really understand this statement. If the authors are trying to say that people may sit for a long time then get up and exercise, thus increasing their mean daily activity, I think it needs to be phrased differently
  • Lines 308-309: So are you saying you looked at their SMARTREAB data and asked them what they were doing at that time? The interview process needs to be described in detail in the Methods section
  • Lines 314-316: “93% of patients… to continuous research.” Please review this sentence. I don’t really follow the logic.
  • Lines 319-326: this is a great case study! See earlier comments about making this more prominent in the results. We also need to understand exactly how these interviews were conducted and how the data was used for activity planning.

Author Response

Response to Reviewer 2 Comments

Santos and colleagues present a descriptive observational study of telemonitoring of daily physical activity in patients with chronic respiratory disease. They use telemonitoring to quantify heart rate, activity, and oxygen saturation over a 4 day monitoring period. They compare this to a validated questionnaire (IPAQ), and determine correlations between the subjective and objective methods. Finally, they describe how the telemonitoring data can be used to create an individualised pulmonary rehabilitation plan.

The study presents an interesting way of integrating objective activity/oxygen monitoring into pulmonary rehabilitation. However, this most interesting part gets lost in a lot of detail and multiple comparisons/correlations. The purpose of the study therefore is a bit unclear. Is this about validating telemonitoring compared to subjective questionnaires? Is this about the pulmonary rehab planning? Due to this lack of clear aims, the ‘novel’ aspects of this study are not really clear to the reader and I think a lot of work needs to be done on this manuscript before it is ready for publication.

General comments:

Point 1: Introduction – there is a lot of detail about findings in each disease category. This level of detail is not necessary to describe the rationale for this study, and all of this information could be summarised in a single paragraph.

Response 1: The authors have reduced and rephrased the Introduction section second paragraph, lines 53-63: “Although 10000 steps/day is a reasonable target for healthy adults who can take between 4000 and 18000 steps/day [3], such values do not apply to patients with chronic respiratory disease. Chronic Obstructive Pulmonary Disease (COPD) patients have less walking time at a slower walk pace [4, 5], with a mean of almost 5000 steps/day [6, 7] and associated increased risks for exacerbation-related hospitalization and premature mortality [8, 9]. On average, all major subtypes of Interstitial Lung Disease (ILD) patients have reduced physical activity with persistent pain and psychological deficits [10], and an annual decline of nearly 1000 steps per day, which is twice as high as compared to COPD patients [11]. About 90% of Bronchiectasis patients do not meet the recommended physical activity guidelines and have more than 10 hours of daily sedentary behavior, walking an average of 6000 steps/day [12]. Daily physical activity is also decreased on Asthma patients with reported 8546 steps/day for mild/moderate Asthma and 6560 steps/day for severe Asthma [13].”.

Point 2: The authors should also make the point here that “average activity level” may not be the most appropriate measurement since there is great variability in the types of activities patients do throughout a day - this is a point the authors later make in the Discussion, and I think is the main rationale for real-time activity/oxygen monitoring.

Response 2: The authors have rephrased the start of third paragraph of Introduction section, lines 64-66: “The average physical activity level measured by the number of daily steps may not be the most appropriate measure to apply as a vital sign of general physical condition in patients with a chronic respiratory disease, given that there is a great variability in patients’ physical activity patterns [5, 14].”.

Point 3: Analysis – there are a lot of correlations made between variables, including categorical variables. While Pearson correlation is appropriate for normally-distributed continuous variables, I am concerned about the use of Spearman correlation for variable that are categorical/ordinal since there will be many “ties” and Spearman rho does not handle this well.

Response 3: Thanks for this comment. The authors have completely revamped and redefined the statistical analyses as described in the Experimental section, lines 165-168:” Associations between categorical variables were analyzed using chi-squared tests or Fisher exact tests, and association between continuous variables weas analyzed with Spearman’s correlation coefficient (r).”, and updated results section. In particular, we replaced most of the correlation previously presented with scatter plots in the supplementary material, as also suggested by the reviewer.

Point 4: We should be able to see the data plotted rather than just r/rho.

Response 4: The authors agree and now present the data plotted in supplementary material and only mention the most relevant findings in the manuscript. Figure S1 is a pair-wise scatter plots in a correlations matrix format, showing the International Physical Activity Questionnaire (IPAQ) total score association with its component subscores. Figure S2 is a pair-wise scatter plots in a correlations matrix format, showing the association between physical activity telemonitoring results and total number of comorbidities or respiratory comorbidities. Figure S3 is a pair-wise scatter plots in a correlations matrix format, showing the association between telemonitoring results of time spent on activities below 2 METs, between 2 and 3 METS, and above 3 METS with International Physical Activity Questionnaire (IPAQ) subscores.

Point 5: Furthermore, if the aim was to validate the SMARTREAB monitoring device against the IPAQ questionnaire, there are probably more appropriate methods than multiple pairwise correlation testing.

Response 5: The authors did not aim to validate SMARTREAB. TELEMOLD preceded SMARTREAB and it was validated as an objective assessment technology by Faria, I.;  Gaspar, C.;  Zamith, M.;  Matias, I.;  das Neves, R. C.;  Rodrigues, F.; Barbara, C., TELEMOLD project: oximetry and exercise telemonitoring to improve long-term oxygen therapy. Telemed J E Health 2014, 20 (7), 626-32., which is reference number 16 on the list. SMARTREAB is an upgraded version of TELEMOLD, maintaining core technology and algorithms of previously validated method. The present study aims to compare results of a subjective method of assessing self-reported physical activity (IPAQ) with the objective method of telemonitoring physical activity (SMARTREAB) with the major purpose of demonstrating this novel input on designing Pulmonary Rehabilitation. The current phase of the project is not of validation but instead of application in clinical practice, with this study aiming to demonstrate its usefulness and increased accuracy compared with other instruments such as questionnaires. We have made these points clearer in the manuscript.

Point 6: I suggest formal statistical review either through the Journal, or through the authors’ own institution.

Response 6: Professor Ruy Ribeiro, Director of the Biomathematics’ Laboratory from our Faculty of Medicine of University of Lisbon, Portugal joined the authors and conducted a formal statistical review of the manuscript. The analyses in the manuscript have been completely re-worked.

Point 7: Since the most interesting and novel part of this study is how the telemonitoring information is used to direct the rehabilitation goals, I think this should be given more prominence in the Results rather than just being mentioned in the Discussion.

Response 7: The authors have added to the Results section, the subsection entitled “3.5. Results applied to designing patient-tailored Pulmonary Rehabilitation programs” in line 310. A new Figure 4 represents SMARTREAB rationale with the example of two clinical cases: a female patient with COPD that only desaturated during the final 10 minutes of 2 hours gardening, and the case of a male patient with Bronchiectasis that had desaturation periods during family support routines. This Figure presents the rationale to operationalize single-patient PADL telemonitoring data as an input to set individually-tailored goals of Pulmonary Rehabilitation, pointing out 2 case examples as suggested. The authors absolutely agree that the most interesting and novel part of this study is how the telemonitoring information is used to direct the rehabilitation goals, and because of that have changed the manuscript title to emphasize this aspect: “Novel input for designing patient-tailored Pulmonary Rehabilitation: Telemonitoring physical activity as a vital sign – SMARTREAB study.”.

Point 8: The case study given is excellent. It would be great to see a second or even third case study – these could be presented as a “box” in the paper (i.e. a separate panel of text, treated like a figure rather than a table – the Journal may be able to provide examples from recently-published articles).

Response 8: The authors have added Figure 4 on the Results section, line 320, presenting 2 case examples as suggested. Authors have created a schematic figure presenting SMARTREAB rationale to operationalize single-patient PADL telemonitoring data as an input to set individually-tailored goals of Pulmonary Rehabilitation.

Specific comments:

Point 9: Abstract, line 28: change “previously” to “prior”

Response 9: The authors have changed the expression “previously” to “prior” on the Abstract section, lines 29-30:” Therefore, it should be clinically assessed as a vital sign of patients’ general physical condition prior to any exercise-based intervention.”.

Point 10: Introduction line 48: “advocates stakeholders for action” this doesn’t really make sense.

Response 10: The authors have eliminated part of the sentence and rephrased the Introduction section, lines 49-52: “Acknowledging this, the initiative Exercise is Medicine® encourages all health care professionals to promptly assess physical activity habits as a vital sign of general physical condition, optimizing exercise counselling or referral and enhancing chronic disease management [2].”.

Point 11: Introduction lines 52-70: these 4 paragraphs should be condensed into one, describing the main point that physical inactivity is common in chronic respiratory disease and is associated with poor outcomes.

Response 11: The authors have rephrased and condensed the Introduction section, second paragraph, lines 55-63: “Chronic Obstructive Pulmonary Disease (COPD) patients have less walking time at a slower walk pace [4, 5], with a mean of almost 5000 steps/day [6, 7] and associated increased risks for exacerbation-related hospitalization and premature mortality [8, 9]. On average, all major subtypes of Interstitial Lung Disease (ILD) patients have reduced physical activity with persistent pain and psychological deficits [10], and an annual decline of nearly 1000 steps per day, which is twice as high as compared to COPD patients [11]. About 90% of Bronchiectasis patients do not meet the recommended physical activity guidelines and have more than 10 hours of daily sedentary behavior, walking an average of 6000 steps/day [12]. Daily physical activity is also decreased on Asthma patients with reported 8546 steps/day for mild/moderate Asthma and 6560 steps/day for severe Asthma [13].”.

Point 12: Line 56: the number of steps/day in COPD is meaningless without a comparison to healthy controls

Response 12: The authors have added this information in the Introduction section, lines 53-55: “Although 10000 steps/day is a reasonable target for healthy adult population that can take between 4000 and 18000 steps/day [3], such values do not apply to patients with chronic respiratory disease.”. For this purpose reference number 3 was added to reference list: Tudor-Locke C, Craig CL, Brown WJ, Clemes SA, De Cocker K, Giles-Corti B, Hatano Y, Inoue S, Matsudo SM, Mutrie N, Oppert JM, Rowe DA, Schmidt MD, Schofield GM, Spence JC, Teixeira PJ, Tully MA, Blair SN. How many steps/day are enough? For adults. Int J Behav Nutr Phys Act. 2011, 8:79.

Point 13: Line 80: “escalated to SMARTREAB” – this doesn’t really describe how it is different to the TELEMOLD study. Is it truly escalated? Or just extended for a different indication/purpose?

Response 13: The authors have changed the expression “escalated” to “upgraded” and rephrased the Introduction section, lines 74-78: “Currently the project has been upgraded to SMARTREAB with physical activity in daily life (PADL) being measured as a vital sign of patients’ general physical condition. Used in clinical routine with chronic respiratory patients, the crossmatching of a combined oximetry and accelerometry with patient qualitative input has been important on designing an individually-tailored Pulmonary Rehabilitation.”.

Point 14: Line 86: aim 2) isn’t really an experimental aim. It should be rephrased as a description only.

Response 14: The authors have rephrased the last paragraph presenting the study aims on the Introduction section, lines 79-81: “The aims of this study were: 1) to compare PADL subjective and objective measures in patients with chronic respiratory diseases; and 2) to demonstrate how SMARTREAB provides important input for designing patient-tailored Pulmonary Rehabilitation.”.

Point 15: Experimental section, line 91: how were the patients selected? Were all patients approached at the time of referral? We should see the number approached, number refused, etc in the results

Response 15: The authors have added required information on the Experimental section, lines 88-92: “Recruitment included 127 people, with 22 patients declining to participate due to personal discomfort, public embarrassment or difficulties in dealing with a smartphone and remote technologies. There were 5 patients who consented to participate but were discontinued from the study given practical difficulties to address the equipment and related poor quality of the telemonitoring output data.”.

Point 16: Line 129: briefly describe how MET is calculated from the telemonitoring data

Response 16: The authors added to the Experimental section lines 123-130 clarifying readers of such matter: “The method for using the triaxial accelerometer data to calculate energy expenditure by METS uses the coefficient of variation (CV) of counts of movement per 10 seconds over a 1-minute period at patient’s waist level. It applies a 2-regression model with an exponential curve for activities like walking or running where CV ≤ 10, and a cubic curve for everyday activities where CV > 10, as fully described by Crouter [18]. Daily activities have a more erratic movement pattern, resulting in greater variability in counts over time, with the model also adjusted by a linear interpolation for activities between 1 and 2.4 METS.” The mentioned algorithms are fully described at Crouter SE, Clowers KG, Bassett DR Jr. A novel method for using accelerometer data to predict energy expenditure. J Appl Physiol 2006, 100 (4), 1324-1331. This last reference was also added as number 18 of list of references.

Point 17: Line 143: how did you conduct this interview? Did the patients keep a diary at the time? How long after the monitoring did you interview them?

Response 17: The authors added to the Experimental section lines 154-157: “Within one to three days after telemonitoring, patients would come to the hospital to crossmatch their written diary with the transcutaneous oxygen desaturation periods (SpO2 < 90%) manually identified by the physiotherapist.”.

Point 18: Lines 143-145: the authors should describe how the information was used to develop their rehabilitation goals – who looked at the data, how did they break it down into relevant periods etc (this is where the “box” case studies would be helpful)

Response 18: The authors added to the Experimental section lines 157-161: “Details about each oxygen desaturation period were recorded in SMARTREAB with patient´s information about: type of activity, intensity, duration, posture, respiratory symptoms and perceived exertion. Data were analyzed and interpreted with patient active involvement in major conclusions, leading to a team definition of goals and design of a Pulmonary Rehabilitation program.”. The Results section, with newly added Figure 4 on line 320, extends clarifying SMARTREAB rationale to operationalize single-patient PADL telemonitoring data as an input to set individually-tailored goals of Pulmonary Rehabilitation, and presents 2 case examples as suggested.

Point 19: Results, line 177: Table 1 should include the number of comorbidities.

Response 19: The authors have added information with median and interquartile range of number of comorbidities per chronic respiratory disease on Table 1 on the Results section, on line 193, after smoking status and before long-term oxygen therapy rows.

Point 20: It is also customary to report “n (%)” rather than “% (n)”.

Response 20: The authors have changed all reported information from %(n) to n(%) on Results section, on Table 1 line 193 and also on Table 2 line 209.

Point 21: “Professional status” is better described as “Employment status”

Response 21: The authors have changed the expression “professional status” to “employment status” on Results section, on Table 1 line 193.

Point 22: Line 186: “medium” should be “median”

Response 22: The authors have changed the expression “medium” to “median” on Results section, line 203.

Point 23: Line 193 Table 2: report “n (%)”.

Response 23: The authors have changed all reported information from %(n) to n(%) on Results section, on Table 1 line 193 and also on Table 2 line 209.

Point 24: Does P25%, P50% etc represent percentiles? If so, please clarify in the table caption.

Response 24: P25%, P50% and P75% represent first, second and third quartiles. Authors have clarified such matter in Table 2 caption, in the Results section, line 211.

Point 25: Line 199: “correlation with smoking” what is the smoking variable? Current, pack years, etc?

Response 25: This has been re-done. The association is with the variable smoking status. The sentence in the Results section now reads, lines 218-221: “There was an association with smoking status (p=0.02), in which active smokers demonstrated less activity than non-smokers, and an association with long-term oxygen therapy (p=0.005), in which patients who use oxygen had lower levels of physical activity.”, reporting to the variable as presented on Table 1, Results section, on line 193.

Point 26: Line 200: “oxygen therapy” is this use of LTOT yes/no?

Response 26: The authors have rephrased the sentence on the Results section, line 218-221: “There was an association with smoking status (p=0.02), in which active smokers demonstrated less activity than non-smokers, and an association with long-term oxygen therapy (p=0.005), in which patients who use oxygen had lower levels of physical activity.”.

Point 27: Line 207: “High PADL by IPAQ total score was associated…” What does this mean? What are the variable you are comparing?

Response 27: As a result of the formal statistical review taken, authors have eliminated that paragraph and rewrote the lines 228-236: “When PADL was analyzed in terms of IPAQ score (MET.min/week), the COPD group had the lowest score, and more than 75% of patients in this group reported physical activity levels below 3000MET.min/week. Regarding patients with Bronchiectasis, 25% had less than 600MET.min/week and more than 25% surpassed 6000MET.min/week, which means there is a wide range of reported physical activity on this group. The associations between IPAQ total score and its component subscores are presented in supplementary online material (Figure S1). Global reported physical activity energy expenditure was positively associated with moderate activity (=0.714, p<0.0005), and this level of activity was strongly correlated with reported gardening and domestic activity energy expenditure (=0.879, p<0.0005).”

Point 28: Line 214: “The previous questionnaire-reported values had a combined analysis with a 4 daytime…” I don’t understand this sentence/analysis.

Response 28: The authors have eliminated this sentence from the Results section.

Point 29: Line 228: “…including the above mentioned sitting time” I don’t see any data on sitting time.

Response 29: The authors have clarified that the sitting time was related to previously mentioned IPAQ result, on the Results section, on lines 256-257:” Results show that patients with COPD, ILD, Asthma and Bronchiectasis spent about 80% of their daytime on activities below 2 METs, including the IPAQ reported sitting time.”.

Point 30: Line 232: there are a number of mentions of “respiratory comorbidities” but this has not been defined

Response 30: The authors have rephrased comorbidities characterization on the Results section, on lines 181-184: “Patients reported a median of 5 comorbidities, and the 3 most frequent were: Arterial Hypertension (56%), Dyslipidemia (40%) and Anxiety or Depression (29%). Along with patient major Chronic Respiratory Disease, the most frequent respiratory comorbidities were: Bronchiectasis (28%), Chronic Respiratory Failure (26%) and Pulmonary Fibrosis (20%).”. Respiratory comorbidities were present along with major Chronic Respiratory Disease (per example: COPD patient with Bronchiectasis and Tuberculosis sequelae – the main diagnostic for attending the Pulmonary Rehabilitation was the COPD disease, and Bronchiectasis and Tuberculosis sequelae were respiratory comorbidities also to consider on patient intervention to be planned).

Point 31: Lines 236-239: What is the purpose of this analysis? I don’t think it is very meaningful

Response 31: As a result of the formal statistical review taken, authors have eliminated this section and rewrote the lines 258-260: “The associations between physical activity telemonitoring results with total of comorbidities or respiratory comorbidities are presented in supplementary online material (Figure S2). There was a large variability in level of physical activity among people with different number of comorbidities.”.

Point 32: Line 240: “…daily oxygen desaturation episodes” how was a desaturation episode defined?

Response 32: The authors have clarified the definition of oxygen desaturation periods on the Experimental section, line 153:” oxygen desaturation periods (SpO2 < 90%)”, and also at Results section, on lines 245-247: “Percentage of SMARTREAB telemonitoring time is presented on Table 3, discriminating PADL intensity by chronic respiratory condition, MET category and SpO2 below 90% (oxygen desaturation period).”.

Point 33: Line 250 Table 4: If I understand this correctly, this is a stratified analysis by IPAQ category? If so, this should be stated so the reader understands. The N for each IPAQ category should be shown in the table.

Response 33: The authors have stated as suggested on Results section, lines 270-271: “as shown on Table 4 with a stratified analysis by IPAQ category”. Authors have also added N for each IPAQ category on Table 4, on Results section, line 273.

Point 34: Discussion lines 279-285: I don’t really see how your data aligns with these patient “clusters”. If the authors believe their data shows something similar, they should explain how/why.

Response 34: The authors have changed the expression “clusters” to “characteristics” on the Discussion section, on lines 341-344: “These patient characteristics are aligned with an interesting published framework analyzing physical capacity and physical activity as a quadrant concept which identified: 34% of patients that “can’t do, don’t do”, 31% of patients that “can do, do do”, 21% of patients that “can’t do, do do” and 14% of patients that “can do, don’t do”[23].”.

Point 35: Lines 298-299: “…as it may be a product of a great variability on patients’ daily physical activity patterns” I don’t really understand this statement. If the authors are trying to say that people may sit for a long time then get up and exercise, thus increasing their mean daily activity, I think it needs to be phrased differently

Response 35: The authors have changed the expression “as it may be a product” to “as it may cover” on the Discussion section, on lines 359-361: “Such decreased physical activity was consistent among all chronic respiratory conditions and it should not be disregarded, as it may cover a great variability on patients’ daily physical activity patterns.”.

Point 36: Lines 308-309: So are you saying you looked at their SMARTREAB data and asked them what they were doing at that time? The interview process needs to be described in detail in the Methods section.

Response 36: The authors added to the Experimental section lines 151-156: “Within one to three days after telemonitoring, patients would come to the hospital to crossmatch their written diary with the transcutaneous oxygen desaturation periods (SpO2 < 90%) manually identified by the physiotherapist. Details about each oxygen desaturation period were recorded in SMARTREAB with patient´s information about: type of activity, intensity, duration, posture, respiratory symptoms and perceived exertion.”. The Results section, with newly added Figure 4 on line 320, extends clarifying SMARTREAB rationale to operationalize single-patient PADL telemonitoring data as an input to set individually-tailored goals of Pulmonary Rehabilitation, and presents 2 case examples as suggested. The Discussion section was rephrased in lines 376-380: ” Considering all this, each of the 100 patients was involved in analyzing and interpreting telemonitoring data, contributing to single-case major conclusions, and committing to individual goals aligned with the design of Pulmonary Rehabilitation intervention. Patient involvement and benefit was achieved, with high satisfaction and impact on daily life, successfully accomplishing Pulmonary Rehabilitation purpose.”.

Point 37: Lines 314-316: “93% of patients… to continuous research.” Please review this sentence. I don’t really follow the logic.

Response 37: The authors rephrased the sentence on Discussion section, lines 372-375: “Such methodology opportunity is crucial for patients’ security, as SMARTREAB showed 93% of patients with daily desaturation episodes. This finding was unobserved by usual clinical setting assessment, revealing a pressing need and interest for continuous research.”.

Point 38: Lines 319-326: this is a great case study! See earlier comments about making this more prominent in the results. We also need to understand exactly how these interviews were conducted and how the data was used for activity planning.

Response 38: The authors have added to the Results section, the subsection entitled “3.5. Results applied to designing patient-tailored Pulmonary Rehabilitation programs” in line 310,  and have reallocated the information previously on the Discussion section to such subsection, in lines 311-318: “SMARTREAB was applied as an innovative telemedicine solution with significant patient involvement in telemonitoring data analysis and interpretation, contributing to single-case major conclusions and personal rehabilitation goals commitment. This means that such process was unique for each patient and could be applied to design individually-tailored Pulmonary Rehabilitation. Figure 4 represents SMARTREAB rationale with the example of two clinical cases: a female patient with COPD that only desaturated during the final 10 minutes of 2 hours gardening, and the case of a male patient with Bronchiectasis that had desaturation periods during family support routines.”. Such part of the text is also supported by newly added Figure 4 on line 320, presenting SMARTREAB rationale to operationalize single-patient PADL telemonitoring data as an input to set individually-tailored goals of Pulmonary Rehabilitation, pointing out 2 case examples as suggested. The authors also rephrased the manuscript title to “Novel input for designing patient-tailored Pulmonary Rehabilitation: Telemonitoring physical activity as a vital sign – SMARTREAB study.”.

Reviewer 3 Report

jcm-850048 telemonitor physical activity respiratory patients

Overall: This paper reports on a study to assess correlation between subjective assessments of physical activity and those measures using a smartphone app for respiratory patients.

  • A thorough review of the grammar is required.

Abstract: Fine as written.

  • Line 26: Not always a consequence as some people were not physically active before being afflicted & not all become inactive.

Introduction: Overall fine in terms of summarizing the problem area, gaps, and thus the rationale for this study.

Methods: Some points require clarification.

  • Line 91: Selected by whom?
  • Line 92: Did you assess reasons for declining?
  • Line 98: What is meant by “openly”? Seems like you just used the survey.
  • Line 98: When were they asked all these questions – prior to using the app?
  • Line 127: Why for only 4 days?
  • Line 130: Can the patient see the data or just the providers?
  • Line 140 & throughout: Data were not data was as data is plural and datum singular.
  • Line 140: Are the data encrypted during sending?

Results: Overall fine with appropriate analyses.

  • Line 208 & throughout: There is no such thing as p = 0.000 – use p < 0.0001 instead.

Conclusions: Overall fine in terms of summarizing the results & discussing in the broader context of the literature.

  • Limitations are noted.
  • Possibly add some thoughts on implications. Since people seem to overestimate their activity times what do you suggest providers tell them when prescribing use of this sort of objective measurement tool – do they trust the device or not? Do they need to really be more active?

References: Fine.

Tables & Figures: Fine.

Author Response

Response to Reviewer 3 Comments

jcm-850048 telemonitor physical activity respiratory patients

Overall: This paper reports on a study to assess correlation between subjective assessments of physical activity and those measures using a smartphone app for respiratory patients.

A thorough review of the grammar is required.

Point 1: Abstract: Fine as written. Line 26: Not always a consequence as some people were not physically active before being afflicted & not all become inactive.

Response 1: Authors rephrased lines 28-29: “Physical inactivity may be a consequence of chronic diseases but also a potential modifiable risk factor.”.

Introduction: Overall fine in terms of summarizing the problem area, gaps, and thus the rationale for this study.

Point 2: Methods: Some points require clarification. Line 91: Selected by whom?

Response 2: Authors clarify in lines 83-85: “Patients with various chronic respiratory diseases referenced for Pulmonary Rehabilitation in Hospital Pulido Valente in Lisbon, Portugal, were invited to participate in the study by CDS between January of 2017 and October 2019.”

Point 3: Line 92: Did you assess reasons for declining?

Response 3: This information is now included in the Experimental section, lines 88-90: “Recruitment included 127 people, with 22 patients declining to participate due to personal discomfort, public embarrassment or difficulties in dealing with a smartphone and remote technologies.”.

Point 4: Line 98: What is meant by “openly”? Seems like you just used the survey.

Response 4: Authors rephrased the sentence in Experimental section, lines 96-98: “Patients were questioned about current physical activity habits (none, or if some on what frequency and modality) and the International Physical Activity Questionnaire [15] (IPAQ) was applied.”

Point 5: Line 98: When were they asked all these questions – prior to using the app?

Response 5: Authors have added new starting expressions on Experimental section paragraphs to clarify that physical activity daily life subjective assessment was prior to objective assessment. Line 96:”The first step was the subjective PADL assessment.”; and line 107: “The second step was the objective PADL assessment.”.

Point 6: Line 127: Why for only 4 days?

Response 6: Authors have added previously mentioned reference 4 in Experimental section line 132: Watz, H.;  Pitta, F.;  Rochester, C. L.;  Garcia-Aymerich, J.;  ZuWallack, R.;  Troosters, T.;  Vaes, A. W.;  Puhan, M. A.;  Jehn, M.;  Polkey, M. I.;  Vogiatzis, I.;  Clini, E. M.;  Toth, M.;  Gimeno-Santos, E.;  Waschki, B.;  Esteban, C.;  Hayot, M.;  Casaburi, R.;  Porszasz, J.;  McAuley, E.;  Singh, S. J.;  Langer, D.;  Wouters, E. F.;  Magnussen, H.; Spruit, M. A., An official European Respiratory Society statement on physical activity in COPD. Eur Respir J 2014, 44 (6), 1521-37. This reference states that “For measurements that aim to assess longitudinal changes 4 days were shown to be sufficient to demonstrate treatment effects following pulmonary rehabilitation in moderate-to-severe COPD”.

Point 7: Line 130: Can the patient see the data or just the providers?

Response 7: As authors have described on Figure 1 caption from Experimental section, lines 114-116: “During telemonitoring patients have access to the smartphone screen information indicating % of transcutaneous oxygen saturation, bpm of heart rate and MET of physical activity.”. To emphasize such information to the reader, authors have added the same information on Experimental section, lines 134-135: “Heart rate, transcutaneous oxygen saturation (SpO2) and MET were monitored, with patient´s access to instant data on the smartphone screen.”.

Point 8: Line 140 & throughout: Data were not data was as data is plural and datum singular.

Response 8: Authors have updated the manuscript with the suggested correction on Experimental section, lines 148-149: “Data collected were processed and sent by GSM/3G/4G to the hospital database server with restricted accessibility by security password associated to different user privileges.”; and lines 156-157: “Data were analyzed and interpreted with patient active involvement in major conclusions, leading to a team definition of goals and design of a Pulmonary Rehabilitation program.”.

Point 9: Line 140: Are the data encrypted during sending?

Response 9: A proprietary communication algorithm, that makes it hard to capture but does not include encryption is used. The hospital database server has restricted accessibility by security password associated to different user privileges, as described on Experimental section, lines 148-149.

Results: Overall fine with appropriate analyses.

Point 10: Line 208 & throughout: There is no such thing as p = 0.000 – use p < 0.0001 instead.

Response 10: The statistical analyses were completely revised and this mistake was corrected everywhere.

Conclusions: Overall fine in terms of summarizing the results & discussing in the broader context of the literature.

Limitations are noted.

Point 11: Possibly add some thoughts on implications. Since people seem to overestimate their activity times what do you suggest providers tell them when prescribing use of this sort of objective measurement tool – do they trust the device or not?

Response 11: The experience with these 100 patients was that people found to be interesting the evidence of quantitative data on the curves of the remote telemonitoring to be a perfect match with activity episodes as they have recorded at their own activity diary. Such information came as a unique new assessment in daily life situation that took place. Patients tended to be surprised with transcutaneous oxygen saturation levels sometimes, especially with these first-time baseline assessments, there was no case of scepticism towards the equipment or the technology and patient trusted the device to be reliable. As IPAQ assessment was prior to SMARTREAB, when it came to the crossmatching phase patients were focused on the telemonitoring data and their activity diary, and no comparison to previously self-reported physical activity was mentioned.

Point 12: Do they need to really be more active?

Response 12: These 100 patients were referred for a Pulmonary Rehabilitation program and being physically active is an established clinical goal to protect health status, decreasing the odds of an exacerbation with need for hospitalization and also decreasing morbidity severity and mortality risk. Our results show that at baseline, these patients are sedentary people before starting Pulmonary Rehabilitation. Authors did not evaluate if beforehand patients felt the need to become physically more active. It is our clinical experience that patients discover how much they were lacking in health status when engaging in increased physical activity habits during the Pulmonary Rehabilitation intervention. 

References: Fine.

Tables & Figures: Fine.

Round 2

Reviewer 1 Report

No more comments

Author Response

Response to Reviewer 1 Comments

No more comments.

Authors have proceeded with minor spell check as required.

Reviewer 2 Report

Thank you to the authors for providing detailed responses to each of my comments. There are only a few outstanding, minor comments to be addressed:

Experimental section: the description of the recruitment is welcomed, however the outcomes of this recruitment should be presented in the results. I suggest a flow diagram to present these results, eg:

PATIENTS APPROACHED (n=127)

--> n=22 declined (reasons)

|

PATIENTS ENROLLED (n=105)

--> 5 discontinued (n due to poor data quality, n due to difficulty with equipment)

|

PATIENTS ANALYZED (n=100)

Line 93 – "latter" should be replaced with "later"

Line 108 – "referenced" should be left as "referred"

Line 155 – "2-regression model", does this mean a model II regression?

Lines 257-264 – the authors do mention the tests they used for comparing categorical variables in the Statistical Analysis section of the Methods, but they should specify in the results whether they used Chi Square or Fisher’s Exact tests for each comparison (I assume Chi square for these particular results given the dimensions of the variables?)

Author Response

Response to Reviewer 2 Comments

Thank you to the authors for providing detailed responses to each of my comments. There are only a few outstanding, minor comments to be addressed:

Point 1: Experimental section: the description of the recruitment is welcomed, however the outcomes of this recruitment should be presented in the results. I suggest a flow diagram to present these results, eg:

PATIENTS APPROACHED (n=127)

--> n=22 declined (reasons)

PATIENTS ENROLLED (n=105)

--> 5 discontinued (n due to poor data quality, n due to difficulty with equipment)

PATIENTS ANALYZED (n=100)

Response 1: Thank you for the comment. The authors have described the recruitment in the Results section, lines 169-170: “Recruitment included 127 people, with 22 patients declining to participate and 5 patients discontinued from the study, as described with detail in Figure 3.”. Such information was removed from the Experimental section. Authors have also presented new Figure 3 with the enrolment flow diagram, as suggested.

Point 2: Line 93 – "latter" should be replaced with "later"

Response 2: Authors have replaced “latter” with “later” in line 73 of the Introduction section.

Point 3: Line 108 – "referenced" should be left as "referred"

Response 3: Authors have replaced “referenced” with “referred” in line 83 of the Experimental section.

Point 4: Line 155 – "2-regression model", does this mean a model II regression?

Response 4: We apply Crouter’s three-part algorithm which is in fact a model with two regressions depending on the CV, with an inactivity threshold to calculate energy expenditure (EE). We use the terminology “2-regression model” to be consistent with the description in Crouter et al.. This is not a model II regression. In detail the algorithm is:

  • if the counts/min are ≤50, EE = 1.0 MET

  • if the counts/min are >50

(2a) and the CV of the counts per 10s are ≤ 10, then EE (METs)   

= 2.379833.[exp(0.00013529.Actigraph counts/min)] (R2 = 0.701; SEE = 0.275)

(2b) or the CV of the counts per 10s are 0 or > 10, then EE (METs)

= 2.330519 + (0.001646. Actigraph counts/min) – [1.2017x10-7 .(Actigraph counts/min)2] + [3.3779x10-12.(Actigraph counts/min)3] (R2 = 0.854; SEE = 0.940)

Point 5: Lines 257-264 – the authors do mention the tests they used for comparing categorical variables in the Statistical Analysis section of the Methods, but they should specify in the results whether they used Chi Square or Fisher’s Exact tests for each comparison (I assume Chi square for these particular results given the dimensions of the variables?)

Response 5: We used Fisher’s exact test to analyse iPAQ categories association with smoking status and with home accessibility, because in these crosstabulations there were categories with less than 5 expected people. We have now explicitly stated this in lines 162-163 of the Experimental section.
